# My experiences with kidney care: A qualitative study of adults in the Northern Territory of Australia living with chronic kidney disease, dialysis and transplantation

Jaquelyne T. Hughes [1,2,3]*, Natasha Freeman[1], Barbara Beaton[3], Anne-Marie Puruntatemeri[4], Monica Hausin[5], Gerarda Tipiloura[6], Pamela Wood[3], Selina Signal[3], Sandawana W. Majoni[1,2,3,7], Alan Cass[1], Louise J. Maple-Brown[1,2], Renae Kirkham[1]

1 Wellbeing and Preventable Chronic Disease Division, Menzies School of Health Research, Charles Darwin University, Darwin, NT, Australia, 2 Division of Medicine, Royal Darwin Hospital, Darwin, NT, Australia, 3 Top-End Renal Service, Darwin, NT, Australia, 4 Elder from Pirlangimpi, Tiwi Islands, NT, Australia, 5 Bilingual speaking renal dialysis recipient, Darwin, NT, Australia, 6 Elder from Wurrumiyanga, Tiwi Islands, NT, Australia, 7 Flinders University and Northern Territory Clinical School, Royal Darwin Hospital Campus, Darwin, NT, Australia

* jaqui.hughes@menzies.edu.au

**Data Availability Statement:** All relevant data are within the manuscript.

## Abstract

### Background

Australian healthcare quality and safety accreditation standards recommend health services partner with health care users, to ensure the highest quality of care. Aboriginal Australians with chronic and end stage kidney disease have high health care access needs.

### Aim

To describe the experiences of health care users of a large government kidney healthcare service provider.

### Methods

Within a government renal health service in the Top-End of the Northern Territory, we undertook a qualitative study involving in-depth interviews with 26 adult clients from urban, regional and remote settings who were living with kidney health conditions.

### Results

Client characteristics included a mean age of 55 years, 55% female and 81% identifying as Aboriginal. The kidney related conditions of client participants included CKD (11, 42.3%), haemodialysis (12, 46.2%), peritoneal dialysis (1, 3.9%), and transplant (2, 7.7%). Key themes emerging from patient interviews related to perceived gaps for clients and carers including: 1) knowledge gaps about the health condition, 2) the impact of relocation in order to access centrally-based renal care, 3) healthcare staff professionalism and qualities and 4) service environments. Overall, the experiences centred on a greater need for client-

**Funding:** The study was funded by Top End Health Services (https://health.nt.gov.au/about/top-end-health-service), who commissioned the work. JH was supported by an Australian National Health and Medical Research Fellowship (1092576). The funders had no role in study design, data collection and analysis, decision to publish, or preparation of the manuscript.

**Competing interests:** JH, BB, PW, SS, SWM, LJMB provided clinical care and/ or were department leaders within the study setting (Top-End Renal Service). AC was a Top End Health Service Board Member during the period of data collection (2017). There are no other relationships or activities that could appear to have influenced the submitted work.The competing interest of authors does not alter our adherence to PLOS ONE policies on sharing data and materials.

centred, respectful and culturally based healthcare support. Clients recommended the need for patient-led collective care, including sustaining an Indigenous Patient Reference Group to support ongoing healthcare service decision processes. Participants included in almost equal proportion, clients with CKD (without dialysis) and clients utilising renal replacement therapy, which adds significant weight to the client-identified recommendations for highest quality of kidney care across a wide spectrum of kidney function.

## Conclusion

Four major themes identified by clients related to their experience with renal care provided by this major regional health care provider: knowledge gaps of their own condition, the lived impacts of relocating to access health care, service environments, and Health Care Provider Quality. An Indigenous Patient Reference Group was one mechanism recommended to support the co-design of preferred care models.

## Introduction

In Australia, the Australian Commission on Safety and Quality in Health Care identifies health care accreditation standards to support optimal care provision and outcomes of care. Partnering with healthcare users to identify ways to enhance care provision is one Australian national clinical quality and safety priority which is recommended to health services [1]. There is also specific guidance for accredited Australian healthcare providers to support engagement and care for First Nations Australian health care users (the Aboriginal and Torres Strait Islander peoples of Australia, or Indigenous Australians) [2].

### Chronic and end stage kidney disease

Chronic kidney disease (CKD) is a term describing sustained impairment of kidney function, and indicated biochemically by low glomerular filtration rate and/ or albuminuria, and early disease is often asymptomatic. Advanced or end stage kidney disease (ESKD), when there is very impaired kidney function, is associated with increasing symptom and requires dialysis and/ or kidney transplantation to sustain life. ESKD treatment is publicly funded in Australia and is provided to more than 25,000 Australians at a cost of more than $1 billion per year.

### Indigenous Australians living in regional and remote settings have high ESKD care needs

Nationally, Aboriginal and Torres Strait Islander peoples (Indigenous Australians) use renal replacement therapy (RRT) to manage ESKD at a 5 times higher rate than other Australians [3], and have lower access to home-based renal replacement therapy including kidney transplantation [4]. The Northern Territory (NT) of Australia is reported to have the highest national prevalence of dialysis dependent ESKD of 2812 v 531 per million population compared to other Australians [5], and the highest incidence of dialysis requiring ESKD among First Nations peoples living across Australian states and Territories at 1344 per million population [3]. In the NT, adults with kidney disease have heavy hospital care needs [6, 7], and recent modelling has projected a linear increase in ESKD in this region over the coming decade [8], which is further supported by evidence of rapid annual decline in kidney function in Aboriginal and Torres Strait Islander adults [9]. However, many Territorians who have the highest

need of ESKD renal care services, live in remote and very remote regions, and are required to relocate over vast distances to access centrally located nurse-assisted-haemodialysis models of care health care services [7, 10, 11]. Advocacy by clients, families, and renal health care clinicians have highlighted this progressive burden of CKD among Indigenous Australians [12], which has been recently supported by nationally targeted policy and funding responses [13].

## A regional renal health care service provider

The Top End Health Services (TEHS) is the NT's largest healthcare provider in the Top-End region. The city of Darwin, with a population of 150,000 persons, is the capital city of the NT, and the operational base of TEHS. Darwin has the smallest population density (44.98 persons per square kilometre) of all Australian capital cities [14]. TEHS provides primary healthcare and hospital care for an urban, regional, remote and very remote living population, and provides comprehensive renal care through Top-End Renal Services [15].

During 2016 the TEHS Integrated Renal Services Review and Action Plan identified the importance of health care user feedback and self-advocacy. TERS lacked processes to receive and respond to feedback, which limited the ability to co-design health care provision with health care users [16]. Furthermore, there were no relevant and validated survey instruments which could be applied to the local health service environment or renal health care user population. Therefore, the Top-End Renal Services Renal Integration Steering Committee invited a partnership with Menzies School of Health Research (involving local clinicians and researchers) to undertake formative qualitative research to describe the range of experiences and satisfaction of renal health care users in the region. As aligned to the business priorities of the 2016–17 TEHS Integration of renal healthcare between primary and specialist care, formative findings of a qualitative research study would then inform the development of healthcare service priorities. In 2017, TEHS with Top-End Renal Services, embarked on a qualitative research study. The aim of this analysis was to describe the experiences of health care users of a large government kidney health service provider.

## Methods

### Participants

Clients were eligible to participate in the research if they had a kidney health condition (general nephrology and CKD, and dialysis and transplantation) and received care from Top End Renal Service. Using a purposive sampling method, which considered the kidney health conditions, and diversity of residential regions (urban, regional and remote areas), potential participants were approached in person to participate by members of the research team through the support of staff at renal clinics. All participants were aged over 18 years old, spoke English, had the study explained to them in English, an interpreter offered, and provided informed written consent.

### The research team

The investigator team for this study included patients, community members, health care providers and researchers with clinical, health services and social science expertise. Of the team, JH is an Indigenous nephrologist and research fellow who co-lead the project in collaboration with RK a non-Indigenous social scientist (PhD in Public Health). BB is a First Nations person from Papua New Guinea and experienced renal services senior Indigenous Liaison Officer. Co-authors, AMP, MH and GT, were members of the Indigenous Reference Group, all of whom speak Aboriginal languages, and have lived experiences of renal replacement care

services. NF is a non-Indigenous researcher with expertise in qualitative research (Masters of Evaluation) who was instrumental in data collection and analysis.

## Interview guides and interview process

Interview guides for study participants are shown in Table 1, and were developed and piloted before undertaking in-depth interviews. Additional contextual perspective was also obtained from carers (as identified by clients). A phenomenological methodology enabled exploration of participants' experiences of accessing healthcare related to kidney health conditions.

Senior researchers (NF and RK) who were unknown to participants led the interviews. Where possible a senior Indigenous Renal Liaison Officer (BB) was present in the proximity of the interviews, to guide a culturally safe interview process for clients. All interviewers were female, with experience in undertaking qualitative research in this context. NF (Masters of Evaluation) and RK (PhD in Public Health) are social scientists who were not involved in the delivery of health care (as some other authors were). In this way, power imbalances between researchers and participants were managed and potential biases accounted for. Interviews were conducted between May—August 2017 in urban, regional and remote locations, in spaces appropriate to both the participant and the interview team including the health centre, their homes or a public space (i.e. park). No individuals invited to participate declined. Interview duration was between 15 to 60 minutes. Demographic and clinical information was collected to provide descriptive information about participants, including age, ethnicity (Indigenous or non-Indigenous), language spoken at home, kidney health condition (CKD without dialysis, haemodialysis, transplantation and peritoneal dialysis) and award of a disability support pension (if relevant).

## Synthesis of interview data

The synthesis and co-interpretation of patient experience was developed in four focus group meetings with Indigenous dialysis clients, in a non-clinical urban setting between April 2017 and November 2017. The development of the focus groups has been described elsewhere [17].

**Table 1. Health care user interview guide.**

| Guiding questions |
| --- |
| *This research wants to understand your experiences and satisfaction with renal services as a renal client to inform future health services planning.* |
| - Can you tell me about your kidney story? |
| - What treatment do you receive? |
| - Are you on any medication? If so, can you explain to me what it is and how often you take it? |
| - Who explained all this to you? |
| - Are you happy with the explanations you receive? |
| - How do you manage your treatment? Transport? Competing priorities such as work and family? |
| - Who supports you with making your appointments and your other commitments? |
| - What can you tell me about the standard of medical care you receive in managing your CKD? |
| - How comfortable do you feel with this treatment? |
| - How comfortable do you feel at the place where you receive your kidney care? (Very comfortable- what makes it comfortable? Not very comfortable- what makes it uncomfortable?) |
| - What, if any, are the challenges associated with your treatment? |
| - Is there anything in particular that you do to look after yourself/your health? |
| - What's your relationship like with the medical/clinic staff who are involved in your treatment? |
| - Who else is supports you with your kidney disease? |
| - How could your treatment be improved/what would you change? |
| - NT Renal Services would like to continue asking how happy you are with the services they provide. What is the best way of providing feedback/ how would you like to them about this/ what is an effective way of giving them this information?' |

Participants of these recurring focus group meetings provided written consent for each focus group to be audio recorded and externally transcribed to further inform findings from this study. These focus group discussions were guided by content raised in previous focus groups and on emerging findings identified from anonymised interviews.

Transcripts from interviews and focus groups were returned to participants who requested this at the time of consent. Transcripts were coded by three members of the research team (NF, BB, RK). NF and RK used NVivo (version 11) to manage data analysis, while BB coded transcripts manually. Data was managed in a secure Excel file. The first round of coding was inductive, to produce new understandings of patient experiences and satisfaction of renal care. The second coding round was deductive based on constructs as directly related to the interview guide, such as satisfaction CKD with renal services (including transport, care received and relationships with staff). The final focus group meeting enabled the accuracy of data interpretation to be cross-checked with participating members. While saturation was reached on many constructs, many contradictory cases were identified which contributed valuable information to this study.

### Governance and ethics

The study was approved by the Department of Health and Menzies School of Health Research Human Research Ethics Committee (HREC 2016–2721). Feedback from early consultation for the study approach recommended establishing an Indigenous Reference Group to guide the data collection, analysis and dissemination phases [17]. Knowledge translation activities included updating service providers and health service clients of the progress of the research through departmental meetings and newsletters.

## Results

Twenty-six clients provided in-depth interview data in the study, including 16 female clients (61.5%) and 21 Indigenous clients (80.8%). Interviews were conducted in the urban, regional and remote settings as follows: 30.8%, 15.4% and 53.8%. The mean age of clients was 55.5 (range 29.5–83.0) years.

The kidney related conditions of client participants included CKD (11, 42.3%), haemodialysis (12, 46.2%), peritoneal dialysis (1, 3.9%), and transplant (2, 7.7%). Eighty six percent of Indigenous clients (18/21) reported their first spoken language was an Indigenous language. Fifty percent of Indigenous clients, and all non-Indigenous clients reported English as the language they spoke most in their home. Eleven clients (42.3%) reported receiving a disability pension, and nine of these clients (81.8%) were receiving maintenance haemodialysis. Contextual information was also received from 1 carer of a client receiving haemodialysis in the urban setting. Saturation was reached on a number of identified themes including limited understandings of CKD, the impact of healthcare professionals and the health service environment, relocation and holistic care needs.

### Knowledge gaps

Clients' understanding of CKD impacted their experiences and satisfaction with renal services. Clients commonly identified kidney health and related management information largely came from doctors, nurses or healthcare clinics. Many clients with CKD were aware of declining kidney function, and all clients recognised the strong family and intergenerational nature of end stage kidney disease in this region. One client described how they understood the cause of CKD:

*"Might be from through family, yeah, my grandpa [...] had a problem, he was on dialysis [...] Goes through the blood."* [7]

Phrasing used by clients to describe declining kidney functioning included having a *"leaky kidney"* or kidneys that *"were getting tired"*. Many clients lacked more detailed understandings of the causes and progression of their own kidney health condition however all were very familiar with "*the [dialysis] machine*".

Communication between the client and health professionals was a key feature of client experience which related to clients' healthcare satisfaction. Poor communication was a recognised barrier to adequate understandings of CKD and optimal care. One third of clients (10 clients) explicitly reported difficulty understanding their doctor or healthcare clinic. The following reasons were identified: complex health language used by staff, rapid speech, unfamiliar speech accents, and insufficient time for a complete explanation. One participant reported that "*...initially nobody talked to me about [transplant]*" which contributed to their dissatisfaction with care.

Clients who were most vulnerable to poor communication included those who felt frightened, had low health-literacy, and were challenged by medical jargon. Clients who felt voiceless in explaining their information challenges to healthcare professionals may offer staff a reassuring response, when this may not be the case:

*"I'm okay [...] rather than [stating the truth which might portray them as] being rude and impolite"* [1]

In contrast, some participants were confident and would articulate for themselves and "speak up" for vulnerable clients with communicating with healthcare professionals. Clients who were confident with communication included long time users of renal health services and spoke English with confidence and fluency. Improvements to the way CKD information was shared were recommended:

*"...I can relate and respond to doctors and nurses and all that. But most of the elderly clients, some of the Indigenous people cannot. They're not literate enough."*

The availability and promotion of communication processes for client feedback and service responses was highly valued:

*"It's good that you mob come down and ask questions to us and we [...] answer back".* [3]

## Health care provider quality

Satisfying health care experiences were linked to receiving support from health staff who conducted care professionally and confidentially and who addressed the client's needs. In addition, clients reported a high degree of satisfaction from their experience of care continuity with staff, and staff that provided technically proficient care.

Clients reported concerns with common dialysis-related procedures, which stemmed from internal factors:

*"I had a line first, because I was frightened (of the fistula) and I had a line. And then after six months they put in a fistula. A line is a catheter that goes in the neck, and that's how they hook up the dialysis".* [14]

Professional and technical competence of staff was important to clients. For example, some participants feared being cared for by staff who had limited experience in their role, and from the anticipated pain or transferred-uncertainty from trainee nurses in dialysis fistula cannulation. Some clients were offended by lack of respect and empathy displayed by some staff. As described by one client who used haemodialysis:

*'I'm starting to get depressed from some of them [. . .] Every time [they say], "Oh this is [client's] last access and we don't want to touch that graft". I said "you're making me depressed. You don't know what that's doing to my brain"'.* [046]

Continuity was highly valued to the health care experience. The positive experience from continuity of care was summarised by one client who stated that *". . .continuity is the key"*. Furthermore, another stated that, *"when there is trust in these relationships, there are more options; more preparedness to work together"* [1]

A lack of continuous care, or ad-hoc care was a source of client dissatisfaction. One client reported frustration with having to repeat his story to different staff, in order to orient staff to his care requirements. This was also important for carers, with one reporting that:

*'You just get to know them and then the next minute they're gone. Then you go to turn around and ask the same question again and half the time the new ones don't really know you or anything'* [12]

### Service environment

Clients reported discontent with transport service inefficiencies to access nurse-assisted haemodialysis in satellite units. Unclear or delayed transportation collection times (of more than one hour) extend a client's time devoted to renal care, which contributes to frustration, post dialysis fatigue, and lack of control. Some felt powerless to assert their rights to a fairer and more transparent process for transport service, yet often had limited resources to create alternative arrangements.

Some clients resided in accommodation services that provided meals and board for a bundled direct debit tariff. When dialysis times and or transport services were delayed or unpredictable, the hostel's food-hygiene policies required hot meals to be discarded. Clients in these accommodations were therefore missing a substantial meal for which they had already paid, but had little option for seeking other meals. Additionally, health hostels did not generally permit cohabitation by a spouse, partner or carer, lacked ensuite bathrooms, and were not functionally suited to clientele with high disability support needs. These factors also contributed to clients' financial concerns of a direct debit bundled tariff when the service was not at contemporary standards.

Within the renal service and hospital environment, several factors contributed to positive or negative experiences. Principally, a culturally respectful environment was valued. Instances of cultural safety being jeopardised were articulated as breaches of privacy, respect and confidentiality, and expressed as "shaming". For example, when on dialysis, confidential issues were sometimes raised in front of staff or other clients in the treatment room. Participants spoke about their preference of using another space for private discussions. In addition, having a welcoming environment in the dialysis unit for family to visit was important to clients:

*'. . .it's good to have family there, sitting by our bedside, but there aren't any chairs for them to sit on'.*

## Carer needs

Despite maintaining a level of independence in personal care, clients indicated support needs additional to care directly related to kidney health. Most had an 'informal' carer to support them as required. Support included: assistance with getting to appointments, someone to talk to, help with domestic duties such as washing and cooking, shopping and providing bush tucker when they were no longer able to collect this for themselves. Usually a family member undertook this '. . . *intensive never-ending job. [My carer is] on patrol, 24 hours, 7 days a week [and] has to be there, like, they are your shadow' [1].*

Clients identified that carers also needed support. Carer pressure was particularly magnified for those with multiple caring responsibilities and carers who had their health concerns. Clients reported that for carers, having to relocate to support family to access dialysis can be disruptive and may require them to relinquish their own jobs, homes, identity and status. Despite this, one carer provided insight into her resilience:

*'You don't worry. . .I just doing things what I got to do.' [12]*

## Impacts of relocation

Many of the dialysis clients interviewed had relocated from a remote community location to an urban or regional centre to access care. For most, the experiences of relocation were negative and included missing family and home:

'*It was very hard for me to leave that place. But I had to you know' [039]*

'*I was crying, I want to go back home'. [4]*

Participants described that following relocation they often relied on family and/or health services to assist them in navigating their new environments. The acceptability and accessibility of accommodation was frequently raised as a factor impacting quality of life. In one instance, an unsatisfactory living arrangement affected a client's medication use:

*'The only time I don't take my medication is when I have a problem with my daughters [. . .] we live in one house. It's all crowded and shouting all day. If I had my own flat I'd be alright'. [037]*

Yet, limitations with other forms of accommodation were commonly described, such as with hostels. A client reported that hostels were a *'compound'* and did not support her idea that *'to improve our condition, we got to live freely.'*

Furthermore, for dialysis clients the importance of being able to return to community to attend events or respond to cultural obligations was frequently mentioned. One client stated the importance that people *'come back [to country, because they] belong to here'.* However, another participant highlighted the complexity of this situation. She was desperately homesick and described how she chose not to take up a respite-visiting haemodialysis chair in her remote community. She believed that if she went home she would not want to leave, and she was not ready for the consequence of ceasing dialysis (she had too much to live for):

*'If I go home for a visit, it will be, "Bye-bye Darwin, I will never come back"'. [14]*

### Client recommendations to enhance the health care service experience

Clients recommended staff to be proficient in cultural awareness and culturally competent within renal care, avoid jargon and account for cultural differences when interacting with clients of different cultural backgrounds:

> '[health professionals] should talk proper so we can understand.'[6]

Clients also recommended *". . .Aboriginal people talking to Aboriginal people, kind of trying to tell the story"*. Client mentor and advocate roles were highlighted as being important, particularly for those new to the renal service, and clients newly transitioned to dialysis care. Clients recognised that client's choices for CKD management were impacted by information sharing strategies. For example, a client described the benefits of their chosen renal care modality with home haemodialysis as providing:

> '. . . me a little bit of freedom, to do it on my own, yeah. Not rely on the nurses'. [33]

This client also noted that more clients should be provided with information about home haemodialysis.

Clients recommended the physical environments in healthcare settings enable both clients and families to feel welcome and comfortable during treatment. An increase in dialysis machines in remote communities was a frequent suggestion, so that clients could reduce the stress and sorrow of dislocation. Recommendations for related service providers for clients with high renal care needs included establishing resident meetings with hostel management, and delivering an efficient transport service (further recommendations are outlined in Table 2).

## Discussion

We present an in-depth qualitative study which explored the experiences and satisfaction with care provided by a major government healthcare provider in the Top-End of the Northern Territory, including for a predominantly Aboriginal health care user clientele. Key factors contributing to the client's healthcare experience included: 1) knowledge gaps about the health condition, 2) the impact of relocation in order to access centrally-based renal care, 3) healthcare provider quality and 4) service environments. Overall, the experiences centred on a greater need for client-centred, respectful and culturally based healthcare support.

### Knowledge gaps

A range of knowledge of the clients' own condition, from low to substantial knowledge was observed, and related to a client's capacity to navigate and self-advocate care needs. Aboriginal people living with chronic kidney disease report knowledge gaps about the condition and care processes [18, 19]. Anderson et al. confirmed the relationship between Indigenous Australians who felt poorly-informed about their CKD and which contributed to confusion and frustration. Client knowledge can be explained by the information sharing strategies, client specific education and learning environments, staff training in adult education, strategies for inter-cultural knowledge sharing, strength based approaches, and utilising key support, such as interpreter services. All of these strategies and processes are beneficial across chronic conditions where Aboriginal peoples are significantly affected [20]. Furthermore, the development of shared understandings and appropriate educational resources are critical to improving client understanding and are a key recommendation from this study [18–21].

## Health care provider quality

Workplace proficiency in renal clinical care and cultural safety were key factors likely to enhance client satisfaction and expectations. Holistic health care delivery contributed to client wellbeing, when the care addressed the clients' medical, social and cultural needs. These care requirements are consistent with holistic care needs identified by Aboriginal and Torres Strait Islander renal care users in separate forums [12]. In this study, we have also observed the association between clients' confidence in their healthcare providers' clinical and cultural competence and their satisfaction with care within the renal service. Clients also valued bi-directional feedback with health caring teams (cached in processes which supported care continuity) [17]. Lack of continuity in care was highlighted, and impacted on progressing and navigating care plans, and has been documented in other health conditions [20, 22].

## An Aboriginal workforce in renal care

Aboriginal clients valued care which was possible with Aboriginal staff. There is low participation of Australian Indigenous peoples in the healthcare workforce within renal services in the

**Table 2. Client identified recommendations to enhance the health care experience.**

|  | Recommendation |
|---|---|
| Knowledge gaps | • Review and use resources and educational material to enhance knowledge sharing<br>• Review and enhance the effectiveness of communication pathways<br>• Support two-way knowledge sharing events between clients and health services teams which enable improved delivery of information about CKD, causes, prevention and treatment options<br>• Enable and ensure family members are included in education sessions<br>• Identify and utilise language interpreters for health care interactions and health care resources and materials |
| Health Care Provider Quality | • Workforce strategy: increased capacity of the Aboriginal Health Workforce, staff capability and professionalism; Involve expert-clients within the formal employment network.<br>• Provide a holistic approach to CKD care/management–inclusive of social and emotional wellbeing.<br>• Enhance opportunities for continuity of care: service response might include Navigators (which could have a regional responsibility).<br>• Health service environment recognising and valuing Indigenous culture in the workplace.<br>• Support Consumer feedback framework, by continuation of the renal client reference group.<br>• Embed dissemination methods to share key policy with clients that have been developed within the reference group. |
| Impacts of relocation and service environment | • Communities need to be informed about the process for transitioning for ESKD care at the time of and following incident dialysis †.<br>• Clients need to be supported to return to communities for ceremonies and other important occasions.<br>• Advocate and enable timely access to appropriate accommodation for clients and families |
| Service Environment | • Enhance environment to be more family friendly.<br>• Transport schedules to be more efficient.<br>• Increase awareness of upholding client confidentiality.<br>• Investigate role of carers further and provide supports for them.<br>• Clients supporting one another through creation of group consultation sessions to assist holistic support and peer support. |

†An inaugural federally funded medicare benefits schedule (MBS) item number to support assisted-haemodialysis in very remote regions of Australia was implemented in November 2018, during the data interpretation and synthesis stage of the study [13].

Northern Territory, and few Aboriginal health professionals across all renal workforce positions [23]. In addition, there is low sustainability of staff in primary health clinics in regions where clients are living with CKD prior to transitioning to dialysis requiring end stage kidney disease [22]. In other settings, a profound and meaningful benefit were observed when clients of underserved and First Nation populations access health care when supported by staff with similar cultural and ethnic background [24, 25]. Expanding the involvement of Aboriginal staff in the renal care workforce was recommended by clients, is consistent with previous recommendations for optimal care [12], is possible [26] and presents an opportunity to improve workplace cultural competence. Clients recognised the low use of health care interpreters in conjunction with confusing jargon, accents, and complex health messages and low health literacy of clients. Interpreters continue to be widely recommended as an appropriate strategy for competent healthcare explanations and also for effective cross-cultural communication [18]. Underuse of interpreters reported in this study is consistent with other studies about health care interpreter uptake and use by healthcare staff reported from this region [27].

## Carer supports

In this study, an enhanced health care experience centred on the strengths of culturally based healthcare support. Where this was lacking, Aboriginal participants described poorer experiences [28]. This is consistent with previously reported patient experiences of separation from family support networks and restricted access to country for family and cultural obligations [21]. Carers provide some cultural and physical support to clients, but carer supports have not been factored into the overarching renal health support matrix. Others have directly advocated that healthcare services strengthening may benefit from supporting carer needs. Associations of heightened ESKD carer burden include carer gender (females), carer poverty, carer health status, and caring burden [29]. A recent systematic review recognised 1) the high burden placed on individuals caring for adults with end stage kidney disease, and 2) limited qualitative understanding of carer needs and supports [30].

## Methodological considerations of identifying and recording renal patient health care experiences

This qualitative study focussed on identifying client healthcare related experiences and satisfaction with care. In this paper, words which we use to describe experiences included: depressed, confusion, frustration, and discontent, powerless, shaming and crying. Some negative experiences were associated with relocation for care accessibility. This study was also not designed to measure or report quality of life, as there was no validated tool for describing the range of client experiences, hence the initiation of this study. There is also no validated quantifiable quality of life measures for Aboriginal patients with CKD in this region who live separated from family and county for care, although these stresses are described by patients [12]. Validating quantifiable measured of emotional wellbeing in this population are recommended, and is underway [31].

The health service recognised the absence of a local and context-specific survey instrument to record and respond to the patient experience. We have identified key themes related to the experience of renal care, and we recommend further research to define a reliable range of responses, and an acceptable and reproducible recording mechanism to suit clients with diverse education, health literacy, and cultural backgrounds and languages. Other survey instruments for renal health have examined psychological health, information needs and opportunities for self-management. We note reporting a survey of activity amongst European renal registries, findings of a systematic review, and views of experts presented at a consensus

forum of health professionals, suggest certain patient-related outcomes measures (PROM) instruments might be appropriate for routine renal registry collection (SF-12, the disease-specific KDQOL-36 and EQ-5D) [32]. It is also recognised, that existing PROM instruments lack validity within Indigenous populations and between non-Indigenous and Indigenous populations, and lack consideration of the holistic and collective health frameworks of Indigenous Australian peoples operating within western models of health care [33, 34]. Health related quality of life and experience measures require validation in Northern Australia for renal care, and we recommend key themes should include provision of clear and useful health information, access to health treatments and culturally safe care [12]. We also note resources are needed (including shared consultation spaces) for health care providers and users to enhance patient-level care decision making [32]. A shared space which allowed recognition and synthesis of these priorities was built within our study governance structure [17], and enabled identification of priorities which sit within and outside of the remit of the health service (health care provider professionalism versus urban and remote accommodation and housing policy). Currently, our national renal registry, ANZDATA, examines patient-level health outcomes, but lacks mechanisms to record and signpost health system policy shifts, individual health service capability, resourcing and workplace culture which may explain patient patient-level health outcomes or experiences. This may be an important Registry innovation.

## Next steps to address these reported experiences

We recommend evaluating the current educational resources and ensuring communication processes are relevant and appropriate for clients and carers. Immediate workforce enhancement may include: 1) identification and use of health care language interpreters to support clients and staff to communicate across language barriers, 2) audit cultural safety proficiency of all renal services staff, and plan for proficiency where lacking, 3) assess workflow units to support care continuity, 4) identify and modify factors that contribute to lack of confidentiality within the treatment spaces and 5) modify the physical environments to support a welcoming environment for clients and their support persons. Both the renal service and participants of the focus group recognised the value of working together to support bi-directional knowledge sharing and feedback, about health services clinical care gaps, innovations and policy. We note TEHS has since formally recognised and operationalised the transition of the study's reference group, into the inaugural the Top End Renal Services Indigenous Patient Reference Group.

## Strengths and limitations

In-depth interviews were an important first stage to informing client experiences measures, however, were demanding of client time and attention. We accept volunteer bias, particularly for clients with competing interests and those who may have felt obliged to participate because of their existing relationship with the research team. However, there was a good representation of clients who lived in and were interviewed in urban, regional and remote settings, and included clients who live with a wide spectrum of kidney health conditions, and many Aboriginal clients participated, which was consistent with the demography of clients with renal care needs within this region [7]. Clients who were unable to participate in this study included patients recently transitioned within 90 days to chronic maintenance haemodialysis [7], and clients who received acute care hospital based. We acknowledge that clients utilising hospital-based dialysis were likely exposed to heightened impact of limited dialysis service capacity [10], and thus our report may underestimate their feedback on care continuity, service quality and transport support. To minimise bias, we established a patient reference group, advertised by expression of interest, which informed and supported a culturally safe engagement model

[17], and which was guided by a respected and trusted senior renal client advocate (BB). The interview guide was piloted and then facilitated by researchers independent of the renal service with expertise in this context. The reported experiences include in almost equal proportion, clients with CKD (without dialysis) and clients utilising renal replacement therapy, which adds significant weight to recommendations for highest quality of kidney care across a wide spectrum of kidney function.

## Conclusion

Four major themes identified by clients related to their experience with renal care provided by this major regional health care provider: knowledge gaps of their own condition, the lived impacts of relocating to access health care, service environments, and Health Care Provider Quality. Clients also reported collective caring preferences, which resonated with patients' perceptions of the missing elements of living well with kidney health conditions. The Patient Reference Group is one mechanism recommended to support the co-design of preferred care models.

## Acknowledgments

We acknowledge the contribution of Mr R Tasman and Ms J Timothy who were members of the study Patient Reference Group who passed away during the period of manuscript preparation. All authors have completed the ICMJE uniform disclosure form at http://www.icmje.org/coi_disclosure.pdf and declare: grant funding of the study from Top End Health Services for the submitted work; Hughes, Beaton, Wood, Signal, Majoni and Maple-Brown provide clinical care and/ or are department leaders within the study setting (Top-End Health Service). There are no other relationships or activities that could appear to have influenced the submitted work.

## Author Contributions

**Conceptualization:** Sandawana W. Majoni, Louise J. Maple-Brown.

**Data curation:** Jaquelyne T. Hughes, Natasha Freeman, Barbara Beaton, Renae Kirkham.

**Formal analysis:** Jaquelyne T. Hughes, Natasha Freeman, Barbara Beaton, Anne-Marie Puruntatemeri, Monica Hausin, Gerarda Tipiloura, Renae Kirkham.

**Funding acquisition:** Jaquelyne T. Hughes, Renae Kirkham.

**Methodology:** Jaquelyne T. Hughes, Barbara Beaton, Anne-Marie Puruntatemeri, Monica Hausin, Gerarda Tipiloura, Pamela Wood, Selina Signal, Sandawana W. Majoni, Alan Cass, Louise J. Maple-Brown, Renae Kirkham.

**Project administration:** Jaquelyne T. Hughes, Natasha Freeman, Barbara Beaton, Renae Kirkham.

**Resources:** Pamela Wood, Selina Signal, Sandawana W. Majoni.

**Supervision:** Jaquelyne T. Hughes, Selina Signal, Louise J. Maple-Brown, Renae Kirkham.

**Validation:** Pamela Wood, Sandawana W. Majoni.

**Visualization:** Jaquelyne T. Hughes.

**Writing – original draft:** Jaquelyne T. Hughes, Renae Kirkham.

**Writing – review & editing:** Jaquelyne T. Hughes, Natasha Freeman, Barbara Beaton, Anne-Marie Puruntatemeri, Monica Hausin, Gerarda Tipiloura, Pamela Wood, Selina Signal, Sandawana W. Majoni, Alan Cass, Louise J. Maple-Brown, Renae Kirkham.

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
