## [Decision Letter · Decision Letter 0]

16 Sep 2019

PONE-D-19-19259

My Experiences with Kidney Care: A Northern Australian qualitative study of adults living with chronic kidney disease, dialysis and transplantation

PLOS ONE

Dear Dr Hughes,

Thank you for submitting your manuscript to PLOS ONE. After careful consideration, we feel that it has merit but does not fully meet PLOS ONE’s publication criteria as it currently stands. Therefore, we invite you to submit a revised version of the manuscript that addresses the points raised during the review process.

ACADEMIC EDITOR: The reviewers raise some valid concerns. I recommend using an appropriate qualitative checklist (e.g. SRQR) to ensure all necessary components of the study are reported.

We would appreciate receiving your revised manuscript by Oct 31 2019 11:59PM. To enhance the reproducibility of your results, we recommend that if applicable you deposit your laboratory protocols in protocols.io, where a protocol can be assigned its own identifier (DOI) such that it can be cited independently in the future. For instructions see: http://journals.plos.org/plosone/s/submission-guidelines#loc-laboratory-protocols

We look forward to receiving your revised manuscript.

Kind regards,

Melissa T. Baysari

Academic Editor

PLOS ONE

Journal Requirements:

2. Please provide additional details regarding participant consent. In the ethics statement in the Methods and online submission information, please ensure that you have specified whether consent was written or verbal/oral. If consent was verbal/oral, please specify: 1) whether the ethics committee approved the verbal/oral consent procedure, 2) why written consent could not be obtained, and 3) how verbal/oral consent was recorded.

JH was supported by an Australia National Health and Medical Research Council Fellowship (1092576).

JH was awarded the funding for the study, which was funded by Top End Health Services (https://health.nt.gov.au/health-governance/top-end-health-service)

The funder commissioned the work, but had had no role in study design, data collection and analysis, decision to publish, or preparation of the manuscript.

This detail is included in the manuscript:

I have read the journal's policy and the authors of this manuscript have the following competing interests: grant funding of the study from Top End Health Services for the submitted work; Hughes, Beaton, Wood, Signal, Majoni and Maple-Brown provide clinical care and/ or are department leaders within the study setting (Top-End Renal Service).  There are no other relationships or activities that could appear to have influenced the submitted work. 

5. Please amend the manuscript submission data (via Edit Submission) to include author Cass A.

Reviewers' comments:

Reviewer's Responses to Questions

**Comments to the Author**

1. Is the manuscript technically sound, and do the data support the conclusions?

Reviewer #1: Yes

Reviewer #2: Partly

2. Has the statistical analysis been performed appropriately and rigorously? 

Reviewer #1: N/A

Reviewer #2: N/A

3. Have the authors made all data underlying the findings in their manuscript fully available?

Reviewer #1: Yes

Reviewer #2: No

4. Is the manuscript presented in an intelligible fashion and written in standard English?

Reviewer #1: Yes

Reviewer #2: Yes

5. Review Comments to the Author

Reviewer #1: Hughes et al report their qualitative work on adults with chronic kidney disease based in Darwin, Australia. The work presented describes their methods and their findings, all of which meet the COREQ checklist.

I have a few minor comments:

- the title suggests northern Australia, yet the work was limited to the top-end of the Northern Territory. Do the authors assume that Torres Strait Islanders would report the same or would a more appropriate title reflect precisely the participant population involved?

- The abstract aim states "To describe the experiences of Aboriginal health care users ...", yet only 81% self-identified as Aboriginal and this specificity was dropped in the final paragraph of the introduction. Could the authors please clarify the aim?

- Selection of participants - was the initial approach made in the participant's language (given 86% of Aboriginal participants had English as second or later language and the researchers were not known to participants).

- How does this work relate to similar findings including patient experience measures? For example the discussion around sharing of knowledge and its relationship to Breckenridge's frameworjk for shared-decision making (Nephrol Dial Transplant 2015, 30: 1605)?

Reviewer #2: This manuscript describes the experiences of Aboriginal health care users of the Top End Health Services' renal services in Northern Territory. Although it seemed to be a service improvement exercise, it is important research and findings will inform policy changes. I nonetheless have a few suggestions for the researchers' considerations;

1. I encourage the researchers to use qualitative reporting checklists like the Standards for Reporting Qualitative Research (SRQR) or the COnsolidated criteria for REporting Qualitative research (COREQ) to guide their writing.

2. A detailed description of the Australian Indigenous community's health and services gaps and, in particular, the situation in the Top End are necessary for context for an international audience.

3. The use of a phenomenological approach, face-to-face interviews and focus groups in this case were appropriate but it would still be useful to know the pros and cons of using this as opposed to other methodologies in an Indigenous population and, in particular again, in the Top End.

4. There should be a description of the credentials, occupations, experience and training of each researcher and their relationships with the participants to give a sense of possible biases and assumptions.

5. There was no mention of software used - coding and analysis were all managed manually?

6. It is not clear exactly HOW the analysis occurred. I read reference no. 11 and still did not fully understand whether the FGs were conducted to co-interpret the interview data OR the FGs were also part of the data that was analysed. I suggest rewriting that paragraph in Methods.

7. Also, the researchers said that the second coding round was based on existing theories but did not say WHICH theories and it is not apparent in Results or Discussion that any theory was used.

Data saturation was not achieved in all themes - there was no description of how this was addressed.

8. Please do not use 'n=...' to describe the proportion of respondents in a qualitative paper.

9. Some of the quotes are quite intriguing but they are not identified (eg. by participant no.) and some of the themes are poorly supported by quotes such as the 'Care and supporting environments' theme.

10. Some of the themes are not consistent with the findings or quotes presented. Eg. Communication and Information - this seems to be more about community understanding of their conditions which may or may not be compounded by communication with and information from health care providers; Commitment and Care Proficiency - this does not describe commitment at all but rather competence of staff and continuity of care; Care and supporting environments - this seems to be describing standards of support and confidentiality. The logic of grouping of these themes are also not apparent to me. I actually thought that the 4 factors described in 1st paragraph in Discussion (knowledge gaps, relocation impact, health care providers' quality, service environments) would serve much better and clearer as themes. I also suggest the use of subthemes to further signpost in the Discussion section.

11. Discussion was fairly well-written. But, again, the use of theories, if any, to interpret and synthesize the findings should be discussed.

12. I think the first few lines in Conclusion should summarize the answer to the aim of this research - what are the experiences of renal patients using TEHS' renal services.

13. The last sentence in Conclusion should really be under strengths and limitations. It isn't a conclusion!

6. PLOS authors have the option to publish the peer review history of their article (what does this mean?). If published, this will include your full peer review and any attached files.

Reviewer #1: No

Reviewer #2: No

---

## [Author Response · Author response to Decision Letter 0]

31 Oct 2019

Thankyou for the opportunity to consider to the recommendations for major revisions, which we have now completed.

---

## [Decision Letter · Decision Letter 1]

12 Nov 2019

My Experiences with Kidney Care: A qualitative study of adults in the Northern Territory of Australia living with chronic kidney disease, dialysis and transplantation

PONE-D-19-19259R1

Dear Dr. Hughes,

We are pleased to inform you that your manuscript has been judged scientifically suitable for publication and will be formally accepted for publication once it complies with all outstanding technical requirements.

With kind regards,

Melissa T. Baysari

Academic Editor

PLOS ONE

Additional Editor Comments (optional):

Reviewers' comments:

Reviewer's Responses to Questions

**Comments to the Author**

1. If the authors have adequately addressed your comments raised in a previous round of review and you feel that this manuscript is now acceptable for publication, you may indicate that here to bypass the “Comments to the Author” section, enter your conflict of interest statement in the “Confidential to Editor” section, and submit your "Accept" recommendation.

Reviewer #2: (No Response)

2. Is the manuscript technically sound, and do the data support the conclusions?

Reviewer #2: Yes

3. Has the statistical analysis been performed appropriately and rigorously? 

Reviewer #2: N/A

4. Have the authors made all data underlying the findings in their manuscript fully available?

Reviewer #2: No

5. Is the manuscript presented in an intelligible fashion and written in standard English?

Reviewer #2: Yes

6. Review Comments to the Author

Reviewer #2: The authors have adequately addressed my comments raised. I recommend 'accept for publication' but do have one suggestion: add a sentence "Strategies were employed to ensure that the methods were respectful to working with Aboriginal and Torres Strait Islander peoples; these included guidance from the Indigenous Reference Group and open-coding involving multiple Indigenosu researchers." to the end of the first paragraph in 'Interview guides and interview process' under methods. Being explicit about what worked and why will serve to inform future research.

7. PLOS authors have the option to publish the peer review history of their article (what does this mean?). If published, this will include your full peer review and any attached files.

Reviewer #2: No

---

## [Editor Report · Acceptance letter]

9 Dec 2019

PONE-D-19-19259R1 

My Experiences with Kidney Care: A qualitative study of adults in the Northern Territory of Australia living with chronic kidney disease, dialysis and transplantation 

Dear Dr. Hughes:

I am pleased to inform you that your manuscript has been deemed suitable for publication in PLOS ONE. Congratulations! Your manuscript is now with our production department. 

With kind regards,

on behalf of

A/Professor Melissa T. Baysari 

Academic Editor

PLOS ONE